# Nurses' Roles in Caring for Older People in Domiciliary Settings: A Scoping Review Protocol

**Isabella Santomauro** [1,*,†], **Erika Bassi** [1,2,†], **Angela Durante** [1], **Consolata Bracco** [3], **Erica Busca** [1,2], **Silvia Caristia** [4] and **Alberto Dal Molin** [1,2]

1   Department of Translational Medicine, University of Piemonte Orientale, Via Solaroli 17, 28100 Novara, Italy; erika.bassi@uniupo.it (E.B.); angela.durante@uniupo.it (A.D.); erica.busca@uniupo.it (E.B.); alberto.dalmolin@med.uniupo.it (A.D.M.)
2   Maggiore della Carità University Hospital, Via Mazzini 18, 28100 Novara, Italy
3   Department of Public Health and Pediatric Sciences, University of Turin, Via Verdi 8, 10124 Turin, Italy; consolata.bracco@unito.it
4   Department for Sustainable Development and Ecological Transition, Via G. Ferraris, 13100 Vercelli, Italy; silvia.caristia@med.uniupo.it
*   Correspondence: isabella.santomauro@uniupo.it
†   These authors contributed equally to this work.

**Abstract:** Due to global shifts in demographics and advances in chronic illness management over the past few decades, domiciliary care has become the primary setting for caring for older people. In this regard, nurses play a crucial role, promoting quality care and minimizing hospital admissions and the need for institutionalization. However, historical and geographic variation in nursing titles and the multitude of labels for different roles have been obstacles to the creation of a clear map outlining specific nursing roles in home care for older people. The aim of this scoping review is to map the evidence on the different nurses' roles in caring for older people in domiciliary settings. This review will include primary, secondary, and gray literature on nurses' roles in domiciliary settings for older people, sourced through comprehensive searches of various databases (MEDLINE, Embase, CINAHL) and reference scanning. No language restrictions will be applied. Two independent reviewers will conduct screening and data extraction. The tabulated results will be informed by descriptive frequencies and content analysis, presenting comprehensive findings. The review protocol was retrospectively registered within OSF database on the 23 November 2023.

**Keywords:** older people; nurse's role; domiciliary settings; primary health care; scoping review

## 1. Introduction

Population aging is a significant and unprecedented phenomenon of the 21st century, with global implications [1]. These are evidenced not only by extended lifespans but also by an increasing percentage of the population over the age of 65. This demographic change brings about a transformation in epidemiological and clinical profiles, with a notable increase in chronic conditions and comorbidities as the population gets older [2]. Consequently, health care systems around the world must urgently develop and implement initiatives that promote healthier living and prevent the need for any kind of institutionalization [3–5].

In recent years, effective approaches to care for older people and the management of chronic diseases have been identified, including the enhancement of Primary Health Care (PHC) services [6,7]. PHC should serve as the population's first and main point of contact with the health system, designed to meet individuals' health needs within the context of their communities and daily lives. The vision of PHC is to be patient-centered, acknowledging the complex nature of health and well-being that encompasses physical, psychological, and social aspects [8]. Domiciliary care, also known as home care, is a key

setting within the PHC framework that provides a range of caring activities to support individuals, particularly older adults, in their own homes [9].

These activities include education, monitoring, personal care, and assistance with routine household tasks, all of which are aimed at enhancing the health and well-being of individuals while maintaining their independence [9]. Currently, home care is considered the most favorable approach for supporting older people, with the aim of preserving their autonomy, as well as their connection to their personal history and identity, for as long as possible [10].

The transition from hospital-based acute care to PHC has prompted a reconfiguration of nursing roles [11,12]. In PHC settings, nurses are crucial in expanding, connecting, and coordinating care [13]. They have proven their capacity to provide safe and effective care across a spectrum that includes disease prevention, diagnosis, treatment, management, and rehabilitation [14]. In particular, nurses in domiciliary care assume a strategic role in managing chronic diseases, promoting self-care, reducing hospital admissions and the need for institutionalization, and enhancing interprofessional collaboration with other health care professionals [15]. These contributions are essential for improving continuity and quality of care.

The literature on PHC nursing employs numerous terms to describe nursing roles, including titles such as 'family health nurse', 'community health nurse', 'district nurse', 'rural nurse', 'nurse consultant', 'nurse clinicians', 'clinical nurse specialist', 'nurse practitioner', 'public health nurse practitioner', 'advanced practice nurse', and 'advanced clinical practitioner' [16–18]. The term 'advanced' is often used to denote the extended nursing roles performed by those with additional qualifications beyond a bachelor's degree [17]. Although there are some landmarks in terms of the education underlying these roles, the plethora of titles can cause confusion about specific competencies and the distinctions between them [19]. This is particularly true in domiciliary care, where it remains unclear which nursing role is best suited to meet the diverse and complex needs of older people in their home environment.

The lack of a clear understanding and consensus on the nature of these nursing roles represents a global challenge, which is further complicated by the differences in nursing education and legislation across various countries. Organizations such as the American Nurses Association (ANA) [20] and the International Council of Nurses (ICN) [21] recognize the absence of a universal definition of nursing roles. In the UK and the US, the roles of clinical nurse specialists and advanced nurse practitioners are encompassed within the scope of advanced practice nursing [22]. In contrast, in Australia, the only regulated advanced role is that of the nurse practitioner [17]. In Canada, PHC nurses, also known as 'family' or 'all-ages' nurse practitioners, represent the most rapidly expanding advanced nursing role in PHC environments [23]. In Italy, a new PHC nursing role known as 'family and community nurse' was established and implemented in 2020 to meet the evolving needs for family and community care, as required by both epidemiological trends and the COVID-19 pandemic [24].

Although the roles of Italian family and community nurses are designed for broader application in various PHC settings [25], they are currently primarily active in the domiciliary care of older people. This responds to the growing demand for senior care and the trend toward home care settings, which are better suited to meet the needs of older people and, in particular, to foster independent and healthy aging.

A preliminary search of PROSPERO, MEDLINE, Open Science Framework, and the Cochrane Database was conducted; no current or ongoing systematic or scoping review mapping the breadth of different nursing roles in caring for older people within domiciliary settings was identified. Some scoping reviews on nursing roles have been conducted, but these were mainly focused on broader PHC settings or addressing specific aspects of the nursing care provided.

The scoping review by Grant et al. examined the practice of PHC nurse practitioners in developed countries and mapped their contribution to improving health outcomes [26].

According to this scoping review, nurse practitioners serving in PHC settings mainly provide care in primary care centers, but also in community centers, outpatient departments, homes, and schools. The extent of their responsibilities spans a spectrum that ranges from focusing on medical conditions to addressing the general health and wellbeing of individuals through a comprehensive approach. Interventions are carried out at the individual and community levels, with outcomes that include increased access to care, cost savings, and salutogenic empowerment for social change.

Another scoping review focused on the barriers and facilitators impacting the implementation of advanced practitioners in PHC [16]. According to the results, the role of the advanced nurse practitioner varies, encompassing care for patients across different age groups and demographics. However, there is minimal consensus regarding the level of autonomy or the definition of the routine tasks of such practitioners. In particular, the 'team factor' emerges as both a barrier and a facilitator in the implementation of new advanced nursing roles within multidisciplinary teams in PHC.

A rapid scoping review by de Leed-Brunsveld et al. focused on mapping studies from northwest Europe that describe and evaluate the community care provided by nurse practitioners within a team of healthcare professionals [27]. According to the authors, nurse practitioners play a key role in meeting the escalating and complex care demands in PHC settings, especially among domiciliary settings.

Currently, there are two ongoing scoping review protocols on nurses' roles in PHC. Sandberg et al. are conducting a scoping review to explore the existing literature on nurses' roles, functions and interventions, including nursing care models, patient care pathways, and clinical practice guidelines, specifically pertaining to older people in long-term care settings [28]. Schlunegger et al. are focusing their scoping review on charting the competencies and scope of practice attributed to nurse practitioners in PHC [29].

Although the majority of the literature on nursing roles to date has focused primarily on broad PHC settings, there has been a recent increase in interest regarding domiciliary care. This change has been prompted not only by an increase in size of the ageing population and chronic conditions, but also by the COVID-19 pandemic, which has emphasized the importance of home-based care alongside the care provided in acute settings [30]. Due to disrupted access to traditional health services, the pandemic further highlighted the necessity—and the possibility—of managing chronic and complex conditions within individuals' homes. This situation has forced governments and policy makers to reconsider how to address people's health needs, redesigning care delivery approaches for the most vulnerable age groups and promoting domiciliary care [10].

Considering the extensive historical and geographic differences in nursing terminology and the growth of diverse nursing roles and titles, this scoping review aims to map the existing literature on nurses' roles in providing care to older people in domiciliary settings.

## 2. Review Question

This scoping review aims to address the following main question: What are the different roles of nurses in caring for older people in domiciliary settings? To comprehensively address this overarching question, the following sub-questions have been identified:

1.   What types of nursing roles are implemented (e.g., family health nurse, community health nurse, district nurse, etc.)?
2.   In what setting do nurses work (e.g., independently, as part of a team)?
3.   Which conditions/health problems do nurses care for (e.g., dementia, heart failure, cancer, etc.)?
4.   What type of caring activities do nurses provide (e.g., health promotion, illness prevention, treatment, rehabilitation)?

## 3. Materials and Methods

The proposed scoping review will be conducted in accordance with the JBI methodology for scoping reviews, considering the PCC framework, where P stands for 'Participants',

C for 'Concept', and C for 'Context' [31]. The review protocol has been registered within the Open Science Framework database: DOI 10.17605/OSF.IO/8NTWD and developed following the guidance provided in the Preferred Reporting Items for Systematic Reviews and Meta-Analysis (PRISMA) extension for protocol (PRISMA-P) [32]. The PRISMA extension for scoping review (PRISMA-ScR) checklist will be used as a guide when reporting the results of this study [33]. A scoping review will be conducted rather than a systematic review due to the broad nature of the research question, which is not aimed at determining the effectiveness or the effects of specific interventions.

### 3.1. Inclusion Criteria

#### 3.1.1. Participants

This scoping review will consider studies that include older people (65+ years). The focus on older adults was a result of their higher frequency of receiving care in domiciliary settings.

#### 3.1.2. Concept

This review will examine the concept of 'nursing roles', specifically focusing on studies that describe the different nursing roles in providing care to older people in domiciliary settings. The roles under consideration will include, but will not be limited to, nurse practitioners, district nurses, advanced practice nurses, clinical nurse specialists, community health nurses, registered nurses, public health nurse practitioners, advanced clinical practitioners, family health nurses, rural nurses, and family nurse practitioners. Due to the lack of a universally accepted definition of nursing roles, the research team adopted an inclusive definition, drawing on elements reported by the ANA and the ICN, which describe a 'nursing role' as the field of action in which nurses act and their distinguishing activities, e.g., the promotion of health, prevention of illnesses, and the care of physically and mentally ill individuals, as well as disabled persons of all ages, in various healthcare and community settings.

Studies will be excluded if they relate to roles such as caregivers, nurse assistants, midwives, and physician assistants.

#### 3.1.3. Context

This review will consider studies conducted in domiciliary care settings. Domiciliary care, also known as home care, comprises a range of caring activities that support people living in their own home. This includes providing education, monitoring, and helping with personal care, all of which are aimed at improving people's health and well-being and maintaining their ability to live independently. Only studies explicitly related to home care settings will be included.

Conversely, studies that focus on nursing homes, assisted living facilities, or facility-based long-term care services will be excluded from this review.

#### 3.1.4. Types of Sources

This scoping review will consider experimental and quasi-experimental study designs, analytical observational studies, and descriptive observational study designs. Qualitative and mixed method studies will also be included. Additionally, secondary studies, e.g., systematic reviews, discussion papers, and policy documents pertinent to the review questions which met the inclusion criteria, will be evaluated for inclusion. Letters and editorials will be excluded. The search strategy will aim to locate both published and unpublished primary studies, reviews, policies, and opinion papers.

#### 3.1.5. Search Strategy

An initial limited search in MEDLINE (PubMed, Bethesda, MD, USA) and EMBASE (Elsevier, Amsterdam, The Netherlands) was carried out to identify articles on the topic of interest. With the support of an expert librarian, the primary author used key terms contained in the titles and abstracts of relevant articles and associated index terms to

develop a complete search strategy for EMBASE (Table 1). The search strategy, including all identified keywords and index terms, will be adapted for each database consulted. Furthermore, the reference lists of all included studies will be screened for additional relevant papers. No language restrictions will be applied. The searched databases will include EMBASE (Elsevier), MEDLINE (PubMed), and CINAHL (EBSCO, Ipswich, MA, USA). Sources of unpublished and grey literature will include ProQuest Dissertation, PROSPERO, as well as governmental and non-governmental documents such as country reports, policies, and regulations related to this topic.

**Table 1.** Search Strategy for EMBASE (Elsevier), conducted on 9 November 2023.

| Search | Query | Records Retrieved |
|---|---|---|
| #1 | ('very elderly'/exp OR 'aged, 80 and over':ti,ab,kw OR 'centenarian':ti,ab,kw OR 'centenarians':ti,ab,kw OR 'nonagenarian':ti,ab,kw OR 'nonagenarians':ti,ab,kw OR 'octogenarian':ti,ab,kw OR 'octogenarians':ti,ab,kw OR 'very elderly':ti,ab,kw OR 'very old':ti,ab,kw OR 'aged'/exp OR 'aged':ti,ab,kw OR 'aged patient':ti,ab,kw OR 'aged people':ti,ab,kw OR 'aged person':ti,ab,kw OR 'aged subject':ti,ab,kw OR 'elderly':ti,ab,kw OR 'elderly patient':ti,ab,kw OR 'elderly people':ti,ab,kw OR 'elderly person':ti,ab,kw OR 'elderly subject':ti,ab,kw OR 'senior citizen':ti,ab,kw OR 'senium':ti,ab,kw OR older:ti,ab) | 6,183,998 |
| #2 | ('nursing role'/exp OR 'nursing role':ti,ab,kw OR 'nurse practitioner'/exp OR 'nurse practitioner':ti,ab,kw OR 'nurse practitioners':ti,ab,kw OR 'practitioner, nurse':ti,ab,kw OR 'district nurse'/exp OR 'district nurse' OR 'advanced practice nurse'/exp OR 'advanced practice nurse':ti,ab,kw OR 'advanced practice nursing'/exp OR 'advanced nursing practice':ti,ab,kw OR 'advanced practice nursing':ti,ab,kw OR 'clinical nurse specialist'/exp OR 'clinical nurse specialist':ti,ab,kw OR 'nurse clinician':ti,ab,kw OR 'nurse clinicians':ti,ab,kw OR 'community nurs *':ti,ab OR 'registered nurse'/exp OR 'registered nurse':ti,ab,kw OR 'public health nurse practitioner*':ti,ab OR 'family nurse practitioner'/exp OR 'family nurse practitioner':ti,ab,kw OR 'family nurse practitioners':ti,ab,kw OR 'family nursing'/exp OR 'family nursing':ti,ab,kw) | 73,268 |
| #3 | ('home care'/exp OR 'domestic health care':ti,ab,kw OR 'domestic healthcare':ti,ab,kw OR 'domiciliary care':ti,ab,kw OR 'domiciliary health care':ti,ab,kw OR 'domiciliary healthcare':ti,ab,kw OR 'home care':ti,ab,kw OR 'home care agencies':ti,ab,kw OR 'home care program':ti,ab,kw OR 'home care programme':ti,ab,kw OR 'home care service':ti,ab,kw OR 'home care services':ti,ab,kw OR 'home care services, hospital-based':ti,ab,kw OR 'home health care':ti,ab,kw OR 'home health nursing':ti,ab,kw OR 'home healthcare':ti,ab,kw OR 'home help':ti,ab,kw OR 'home nursing':ti,ab,kw OR 'home service':ti,ab,kw OR 'home treatment':ti,ab,kw OR 'homecare':ti,ab,kw OR 'homemaker service':ti,ab,kw OR 'homemaker services':ti,ab,kw OR 'hospital-based home care services':ti,ab,kw OR 'domiciliary setting':ti,ab) | 102,723 |
| #4 | #1 AND #2 AND #3 | 894 |

### 3.2. Study Selection and Screening

After conducting the search, all identified citations will be uploaded to Rayyan (http://rayyan.qcri.org (accessed on 12 November 2023)), and duplicates will be removed. Following this, a pilot test covering 10% of the retrieved citations will be conducted to achieve a 75% agreement rate between reviewers. Subsequently, two reviewers will independently screen the titles and abstract to evaluate them against the inclusion criteria.

Potentially relevant papers will be retrieved in full, and their citation details will be imported into Rayyan. The full text of the selected citations, after conducting a pilot test covering 5%, will be evaluated in detail against the inclusion criteria by two independent reviewers. The reasons for exclusion of full-text articles that do not meet the inclusion criteria will be recorded and reported in the scoping review.

Any disagreement that arises between the reviewers at each stage of the selection process will be solved through discussion or with a third reviewer. The results of the search will be fully reported in the final scoping review and presented in a PRISMA flow diagram [34].

### 3.3. Data Extraction

Data will be extracted from articles included in the scoping review by two independent reviewers using a data extraction tool adapted from the JBI framework data extraction instrument [35]. The data extracted will include specific details on the participants, concept, context, study methods, and key findings relevant to the aforementioned questions. A draft extraction tool is provided (Table 2).

**Table 2.** Data extraction instrument.

| **Scoping Review Details** |
| --- |
| Review Title |
| Review Aim/s |
| Review Question/s |
| **Inclusion/Exclusion Criteria** |
| Population |
| Concept |
| Context |
| **Type of evidence source** |
| Evidence source details and characteristics |
| Citation details (e.g., author/s, date, title, journal, volume, issue, pages) |
| Country |
| Participants |
| Context |
| **Details/Results extracted from source of evidence (in relation to the concept of scoping review)** |
| Type of nursing role |
| Type of practice setting |
| Type of health conditions/problems cared for |
| Type of caring activities performed |

The draft data extraction tool will be modified and revised as necessary during the process of extracting data from each included paper. Changes will be enhanced in review. Any disagreement that arises between the reviewers will be resolved through discussion or with a third research member. The authors of articles will be contacted to request missing or additional data, if necessary.

### 3.4. Data Analysis and Presentation

The search process will be presented, along with a study flow diagram. Data supporting the review question will be analyzed and presented in tabulated form using charting techniques. Furthermore, descriptive frequencies, word count, and content analyses will be presented to show how the findings relate to the aims of the review.

For the presentation of the tag 'type of nurse role', a word cloud will be generated. Any gaps in the data will be highlighted so that future researchers can focus on those.

## 4. Contribution to the Topic

By mapping out the varied roles within this specific care setting, this scoping review addresses the lack of a clear framework for nursing roles in home care, compounded by historical and geographic variations in job titles and responsibilities. Examining nursing practices across different regions and periods, the review aims to enrich the global perspective on nursing, encouraging a more unified approach to education and practice. Moreover, by identifying gaps in the current literature, this review could set the stage for future research, targeting under-explored areas and interventions in domiciliary care.

**Author Contributions:** Conceptualization, I.S., E.B. (Erika Bassi), S.C., E.B. (Erica Busca), A.D. and A.D.M.; methodology, I.S., E.B. (Erika Bassi) and A.D.; investigation, I.S., E.B. (Erika Bassi) and A.D.; writing—original draft preparation, I.S. and E.B. (Erika Bassi); writing—review and editing, I.S., E.B. (Erika Bassi), A.D. and C.B.; supervision, A.D.M.; project administration, I.S., E.B. (Erika Bassi) and A.D.M.; funding acquisition, A.D.M. All authors have read and agreed to the published version of the manuscript.

**Funding:** This study is part of the AGE-IT which has received funding from the MUR-M4C2 1.3 of PNRR with grant agreement no. PE0000015.

**Institutional Review Board Statement:** Not applicable.

**Informed Consent Statement:** Not applicable.

**Data Availability Statement:** The data presented in this study are all available in this manuscript.

**Public Involvement Statement:** No public involvement in any aspect of this research.

**Guidelines and Standards Statement:** This manuscript was drafted against the Preferred Reporting Items for Systematic Review and Meta-Analysis (PRISMA) extension for protocol (PRISMA-P).

**Use of Artificial Intelligence:** AI or AI-assisted tools were not used in drafting any aspect of this manuscript.

**Acknowledgments:** Roberta Maorett for support in developing the search strategy.

**Conflicts of Interest:** The authors declare no conflicts of interest.

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
