# Peer review of "Nurses’ Roles in Caring for Older People in Domiciliary Settings: A Scoping Review Protocol"

_nursrep, doi:10.3390/nursrep14020057_

Round 1

Reviewer 1 Report

Comments and Suggestions for Authors

The article presents a SLR conducted by JBI protocol. Good introduction. Study problem explain based on recent evidence. Demonstrates de all stages of the methodology.

Provides sufficient background and include all relevant references to explain the study problem.  Pay attention on the congruence of table numeration (table 1, table II).

Comments on the Quality of English Language

Minor editing of English language required.

Reviewer 2 Report

Comments and Suggestions for Authors

This manuscript presents: Nurses’ roles in caring for older people in domiciliary settings: A Scoping Review Protocol. 

I have some comments and concerns, listed below: 

1. Abstract

Long introduction, but no results or conclusions.

2. Introduction

- wrong way of citation (line 121, 124)

- no space between the bibliography and the word throughout the manuscript

3. Material and Methods 

In my opinion section Material and Methods is too vague. 

4. Authors should add more information about aim. 

Reviewer 3 Report

Comments and Suggestions for Authors

Thank you for the opportunity to review this paper "Nurses' roles in caring for older people in domiciliary settings: A scoping Review Protocol." This protocol is well written. One suggestion follows.

The authors may also include the role(s) of nurses in a team practice setting or as one of the team members.

Reviewer 4 Report

Comments and Suggestions for Authors

First of all, congratulations on your work. Please consider the following recommendations as opportunities to improve the quality of the manuscript.

Introduction

Regarding the introduction, this section is well-built and leads to the main review question. However, as a recommendation, consider developing more into the nursing therapies that can be developed within the scope of domiciliary care (paragraphs 57-65). This may indicate the need to map the several approaches within this aspect and enhance the importance of the review.

Portugal also has a model where a nursing specialist exists for community and family care, where domiciliary care is a reality. Although understandable, this may impact your option for limiting the search for articles in English, Italian and Spanish.

Review Question

Consider incorporating a major objective and divide it into sub-questions, as presented in Table II. Consider including sub-questions related to Table II's extraction columns, precisely the type of nursing role, the settings, and the therapeutics developed by the professionals. Also, consider incorporating an example after each question.

Material and methods

Inclusion criteria are well-built. However, as stated within the JBI's recommendations regarding scoping review development:

"Our strong recommendation is that there are no restrictions on source inclusion by language unless there are clear reasons for language restrictions (such as for feasibility reasons)."

Consider using DeepL to translate documents, or provide a proper justification for the language restrictions.

The pilot stage is not defined. Consider indicating the number of articles included within each stage (title/abstract, full-text, extraction). For example, 10% of the title/abstract stage and 5% of the full-text stage will be used until an agreement of 75% is reached. Do the same for the extraction stage.

Overall, the review protocol is well-built but requires these minor enhancements to align with JBI's approach.

Comments on the Quality of English Language

Regarding the methodology, the article should be written in future tense.

A simple review of the English may help in a more fluent and readable. As an example, "Currently, home care at home..." (line 53), is redundant.

Reviewer 5 Report

Comments and Suggestions for Authors

 The article is of particular interest. I think the authors have met the scientific requirements for carrying out this protocol. In order to improve it, I suggest mentioning in the introduction the ways in which the scoping review will advance the nursing discipline.

Round 2

Reviewer 2 Report

Comments and Suggestions for Authors

Dear Authors,

thank you for improving the manuscript.